# Use of Xenogenic Collagen Matrices in Peri-Implant Soft Tissue Volume Augmentation: A Critical Review on the Current Evidence and New Technique Presentation

**DOI:** 10.3390/ma15113937

**Published:** 2022-05-31

**Authors:** Carlo De Annuntiis, Luca Testarelli, Renzo Guarnieri

**Affiliations:** 1Private Perio-Implant Practive, 00100 Rome, Italy; carlodea@hotmail.it; 2Department of Oral and Maxillofacial Sciences, University La Sapienza, 00100 Rome, Italy; luca.testarelli@uniroma.it

**Keywords:** xenogenic collagen matrix, connective tissue graft, RCTs

## Abstract

Plastic peri-implant surgical procedures aiming to increase soft tissue volume around dental implants have long been well-described. These are represented by: pedicle soft tissue grafts (rotational flap procedures and advanced flap procedures) and free soft tissue grafts (epithelialized, also called free gingival graft (FGG), and non-epithelialized, also called, connective tissue graft (CTG) or a combination of both. To bypass the drawback connected with autologous grafts harvesting, xenogenic collagen matrices (XCM)s and collagen-based matrices derived from porcine dermis (PDXCM)s have been introduced, as an alternative, in plastic peri-implant procedures. Aim: This review is aimed to evaluate and to critically analyze the available evidence on the effectiveness of XCMs and PDXCMs in soft tissue volume augmentation around dental implants. Moreover, a clinical case with a new soft tissue grafting procedure technique (Guided Soft Tissue Regeneration, GSTR) is presented. Material and Methods: An electronic search was performed on the MEDLINE database, SCOPUS, Cochrane Library and Web of Science. The electronic search provided a total of 133 articles. One hundred and twenty-eight not meeting the inclusion criteria were excluded. Seven articles of human randomized clinical trials were selected. A total number of 108 patients were treated with CTG, and 110 patients with XCM. Results: in peri-implant soft tissue augmentation procedures, XCMs seem an effective alternative to CTGs, associated with lower patient morbidity and lower operative times.

## 1. Introduction

Soft tissues around implants differ from those around teeth regarding the amount of blood supply, the direction of connective tissue fibers, the number of fibroblasts and collagen fibers, and the permeability of junctional epithelium [1]. To clinically describe the morphologic and dimensional features of soft tissue components around dental implants, recently, it has been introduced in literature the term “peri-implant phenotype” [2,3]. This includes four parameters: (1) the peri-implant keratinized mucosa width (PKMW), (2) the peri-implant soft tissue thickness (PST), (3) the peri-implant supra-crestal tissue height (PSTH), and (4) the peri-implant bone thickness (PBT). The PKMW represents the vertical height of the keratinized gingiva that goes from the free gingival margin to the muco-gingival line [2]. Even if there are conflicting opinions in the literature [4], it is generally accepted that 2 mm of PKWM is needed to maintain optimal bacterial plaque control and limit the risk of mucosal recessions [5,6,7,8,9,10,11,12,13]. The PST represents the thickness of the peri- implant soft tissue. It is measured horizontally at the base of the peri-implant sulcus or at the most coronal part of the implant shoulder [2]. This horizontal measurement, defined in the past as “mid-facial peri-implant mucosa”, has been used for the esthetic evaluation around dental implants [14,15,16,17,18,19]. As for the PKMW, also for this parameter, there is a general consensus that 2 mm of PST is needed to obtain the so called “masking effect “, i.e., to mask the abutment’s coloring [18]. The term PSTH represents the vertical dimension of the peri-implant soft tissue, that goes from the free gingival margin to the crestal bone [2]. In the past this dimension was called “peri-implant biologic width”, being constituted by the sulcular epithelium, the junctional epithelium, and the supra-crestal connective tissue. The PBT is the horizontal dimension of the osseous tissues supporting a dental implant.

Significant evidence supports the importance of the three-dimensional volume stability for over time functional and esthetics outcomes of dental implants [19]. Consequently, in case of inadequate conditions, several peri-implant plastic surgical techniques have been proposed to obtain a soft tissue volume augmentation. These are represented by pedicle soft tissue grafts (rotational flap procedures and advanced flap procedures) and free soft tissue grafts (epithelialized, also called free gingival graft (FGG), and non-epithelialized, also called, connective tissue graft (CTG) or a combination of both) [19]. FGG has been proved the most effective in recreating PKMW, PST and PSTH. Nevertheless, a significant increase of PKMW, PST and PSTH has also been documented with CTG. High patient morbidity, high postoperative discomfort, and high risk of complications are the major side effects of CTGs [19]. To bypass these drawbacks, recently xenogeneic collagen matrices (XCMs) have been proposed as an alternative to the CTGs [16]. Several studies have investigated the clinical outcome of XCMs and PDXCMs in plastic peri-implant surgery. These studies have shown favorable clinical results; however, the available data demonstrate that surgical outcomes might be influenced by many conditions. In this context, the aim of this literature review is to investigate whether the use of XCMs and PDXCMs provides similar outcomes of CTGs in peri-implant soft tissue augmentation procedures and to critically analyze the available evidence. Moreover, a clinical case with a new PDXCM and a new grafting procedure is presented.

## 2. Materials and Methods

An electronic search was performed on the MEDLINE database, through PubMed (www.ncbi.nlm.nih.gov/pubmed, accessed on 15 January 2022), SCOPUS (www.scopus.com, accessed on 15 January 2022), Cochrane Library (www.thecochranelibrary.com, accessed on 15 January 2022) and Web of Science (www.webofknowledge.com, accessed on 15 January 2022) using the following key words connected by the Boolean operators OR, AND: “xenogeneic collagen matrix”, “connective tissue graft” “dental implants”, “soft tissue augmentation”, “soft tissue volume augmentation”. No time restriction was applied. Studies were selected for inclusion if they met the following criteria: (1) Human randomized and prospective clinical trials, (2) surgical treatment aimed at increasing peri-implant soft tissue volume, (3) comparison of CTG (control) versus XCM (test), (4) follow-up of at least 3 months, (5) reported outcomes measures PKMW or PST or PSTH following the surgical intervention. The exclusion criteria were: (1) study with < 10 patients, (2) case-control studies, case series, case report, and systematic reviews, (3) in vitro studies (4) surgical treatment including materials others than autogenous connective tissue or XCM. The following data were extracted from each article: names of the authors, year of publication, study type, description of the sample size, follow-up period and outcomes. In addition, the search was complemented by a manual search of relevant articles published in the following journals: Journal of Oral Rehabilitation, Clinical Oral Implants Research, International Journal of Oral and Maxillofacial Implants, Implant Dentistry, Clinical Implant Dentistry and Related Research, International Journal of Periodontics and Restorative Dentistry, International Journal of Prosthodontics, Journal of Oral and Maxillofacial Surgery, Quintessence International, Journal of Periodontology, International Journal of Oral and Maxillofacial Surgery, Journal of Oral Implantology, and Journal of Clinical Periodontology (Table 1).

Focused questions (based on PICO criteria): What are the clinical effects of XCM (I) relative to soft tissue augmentation methods (C) on improving PKMW, PST and PSTH (O) around dental implants (P)?

In order to increase the quality and transparency of the study, the PRISMA checklist was followed (Table 1).

Finally, the references of all selected full-text articles were searched for relevant articles. In Table 1 are reported the outcomes of the selected clinical trials published following use of XCM vs. CTG. Seven randomized clinical trials were selected including two parallel arms, one treatment arm using CTG and in the other using XCM [20,21,22,23,24,25,26]. The follow-up time of RCTs ranged from 3 to 12 months (mean 6 months). Patients’ characteristics: a total number of 108 patients were treated with CTG, and 110 patients with XCM. All patients were periodontally healthy, smoking < 10 cigarettes/day. In 44 patients [20,21] surgical techniques were performed after implant placement, and in the remaining 64 before crown delivery (Table 2). In addition, in the present critical review, results of two prospective cohort studies on use of XCMs [27,28], and of seven clinical studies on use of PDXCMs were also reported (Table 2, Table 3, Table 4 and Table 5).

### 2.1. Assessments of the Risk of Bias

The risk of bias analysis was performed by one reviewing author (R.G), using the Cochrane Collaboration’s tool for assessing RCT risk of bias. In Table 3, parameters used for analysis of risk of bias (low, medium and high) are reported.

### 2.2. Statistical Analysis

The continuous outcomes were expressed as mean difference (MD), with a confidence interval (CI) of 95%. Chi-square tests were used to assess the heterogeneity of RCT. Values ≤ 25% = low heterogeneity, values > 25 ≤ 50% = moderate heterogeneity, values ≥ 50% = high heterogeneity. The random effect model was used when heterogeneity was found (*p* < 0.10).

## 3. Results

Compared to XCM, the CTG yielded an increment difference in PKMW of 0.19 mm (−0.03, 0.41), but the difference was not statistically significant. The increase in PST was reported by 2 trials: compared to XCM, the CTG yielded an increment difference of 0.07 (−0.39, 0.53), without a statistical difference. Compared to CTC, XCM yielded an overall PKMW gain difference of −0.06, but this was not statistically significant, regardless the surgical technique used (apically positioned flap vs. bilaminar technique). The difference in postsurgical discomfort, evaluated with the visual analog scale was 1.98 (0.63, 3.33) in favor of XCM, with a statistically significance. A significant longer treatment time (15.46 min.) was associated with CTG, compared to XCM (Table 4).

**Table 4 materials-15-03937-t004:** General overview of the results reported by selected RCT which compared CTG (control) versus XCM (test).

General Overview of the Results of RCT Which Compared XCMs (Test) Versus CTGs (Control)				
Study	PAS (VAS on a 0 to 100)	Changes in PKMW between Baseline and Final Follow-Up (mm)	Changes in PD between Baseline and Final Follow-Up (mm)	Comments by Authors
Sanz et al. [20]	N.R	CTG 2.6 ± 0.96 XCM 2.5 ± 0.7	N.R.	The XCM was as effective and predictable as the CTG for attaining a band of keratinized tissue, but its use was associated with a significantly lower patient morbidity.
Lorenzo et al. [21]	N.R	CTG 2.33 ± 1.03 XCM 2.3 ± 0.47	CTG 0 ± 1.03 XCM 0.4 ± 0.62	The results of the study demonstrate that the use of XCM presented similar results to the CTG for the KM band gain.
Thoma et al. [22]	0–10	CTG 0.8 ± 1.8 o. 0.8 ± 2.2 b. 1.6 ± 2.6 a. XCM 1.4 ± 1.4 o. 1.1 ± 1.4 b. 0.9 ± 1.9 a	N.R	The XCM was as effective and predictable as the CTG for attaining a band of keratinized tissue
Zeitner et al. [23]	0–10	CTG 4.2 ± 1.9 o. 4.1 ± 2.0 b. 3.4 ± 1.8 a. XCM 3.4 ± 1.0 o. 2.9 ± 1.5 b. 2.6 ± 2.3 a.	N.R	The use of XCM and the subepithelial connective tissue graft for soft tissue augmentation at implant sites rendered a similar gain in soft tissue volume
Cairo et al. [24]	CTG 90 ± 9.0 XCM 90 ± 8.0	CTG 0.9 ± 1.6 XC 1.2 ± 1.2	CTG 2.9 ± 0.3 XCM 2.8 ± 0.2	Similar gain in keratinized tissue and in the peri-implant soft tissue thickness
Puzio et al. [25]	N R	(change) CTG 1.52 ± 1.0 XCM 0.89 ± 0.6	N R	Both XCM and CTG increase the keratinized tissue but higher values were noted using CTG
Huber et al. [26]	N R	CTG 3.2 ± 0.8 XCM 2.1 ± 1.2	N R	The buccal peri-implant soft tissue dimensions at implant sites revealed only minimal changes without relevant differences between sites that had previously been grafted with XCM or CTG.
**General overview of the results of prospective studies which investigated XCMs.**				
Papi & Pompa [27]	NR	XCM 4.32 ± 1.22	0.38 ± 0.21	With XCM, the keratinized tissue width can be augmented, and the width remains stable for the assessment period of 12 months.
Schallhorn et al. [28]	90 ± 20	XCM 2.1 ± 1.0	3.0 ± 1.6	XCM demonstrated the potential to increase KMW and GT around existing dental implants.

NR = not reported; o = occlusal, b = buccal, a = apical.

## 4. Discussion

Comparative results in PST and PKMW at 6 months have been reported by Cairo et al. [24] who, in a randomized clinical study, performed soft tissue augmentation at 60 implants in 60 patients during implant uncovering. In the CTG group the authors found a final PST increase of 1.2 ± 0.3 while in the XCM group it was 0.9 ± 0.2, with a significant difference (0.3 mm; *p* = 0.0001). However, both procedures resulted in similar final PKMW amount with no significant difference between treatments. Comparative similar results in the final PST increase were indeed obtained in a RCT by Thoma et al. [22], who treated 20 patients obtaining with XCM a mean soft tissue thickness increase at 90 days post-surgery of 1.4 ± 1.4 mm (occlusal) of 1.1 ± 1.4 mm (buccal) and of 0.9 ± 1.9 mm (apical). The corresponding values obtained with CTG were 0.8 ± 1.8 mm, 0.8 ± 2.2 mm and 1.6 ± 2.6 mm, respectively. Sanz et al. [20] performed soft tissue volume augmentation of 20 randomized implants which presented <1 mm of keratinized tissue. Ten patients received CTG, and 10 patients received XCM. At 6 months, the CTG group showed a mean width of keratinized tissue of 2.6 (SD 0.9) mm, while in the XCM group it was 2.5 (SD 0.9) mm, these differences being insignificant. A 60% and 67% of volume contraction was recorded in the CTG and XCM group, respectively, without variations of periodontal parameters between groups. Compared to CTG group, a lower patient morbidity and a reduced surgery time was recorded in the XCM group. Difference in outcomes between the study by Cairo et al. [24], Thoma et al. [22] and Sanz et al. [20] may be linked to the type of XCM used. Unlike what was performed by Cairo et al. [24], who used a double layer XCM, Thomas et al. [22] and Sanz et al. [20] used a three-dimensional stable XCM. The structure of this XCM consists of two functional layers: a cell occlusive layer consisting of collagen fibers in a compact arrangement and a porous layer. The porous layer is thicker in order to achieve more keratinized tissue by inducing a space-creating effect and by favoring blood clot formation. The same XCM was also used in two RCTs by Lorenzo et al. [21] and Huber et al. [26]. Lorenzo et al. [21] at 6 months post-surgery in the CTG group attained a mean PKMW of 2.75 mm, while in the XCM group the mean PKMW was 2.8 mm, the inter-group differences not being statistically significant. Moreover, in both groups a similar esthetic result and a similar significant increase in the vestibular depth as a result of the surgery was recorded. Huber et al. [26] obtained at 1-year post-surgery a mean PST value of 3.0 mm for XCM, and of 2.8 mm for CTG without statistically significant differences within and in between the groups.

None of the RCTs selected by the current review report results relating to PSTH and PBT after treatment with XCMs vs. CTG. In peri-implant soft tissue augmentation procedures, an increase in PSTH should still be within the therapeutic goals as it has been shown that a thickness > 2 mm could have a protective effect on peri-implant marginal bone resorption [2]. In addition, even a minimum thickness of 1.5 mm of PST should always be present to reduce peri-implant bone remodeling [2].

Although results of the selected randomized clinical trial proved that XCM, when used as a soft tissue substitute aiming to increase the peri-implant tissue volume was as effective and predictable as the CTG with a significantly lower patient morbidity, they indicated a higher shrinkage rate of XCM compared to CTG over time. Clinically, three-dimensional tissue alterations following soft tissue thickening around dental implants were quantified in a prospective study by Schmitt et al. [29] using a digital method able to superimpose the baseline model with the one obtained after surgery. The results indicated a contraction of the initially augmented soft tissue volume of 81.76% in the XCM group and 56.39% in the CTG group after 6 months. Histological studies showed that, during healing, CTG is encapsulated [30], while XCM undergoes to remodeling process without encapsulation [31,32]. This could justify the higher shrinkage rate of XCM compared to CTG after soft tissue thickening [33]. Another factor which could favor the higher shrinkage rate of XCMs vs. CTGs is the inflammatory response and the foreign body reaction connected to cross-linking molecules present in XCM [34]. Moreover, the difference in loss of volume between groups treated with CTG vs. XCM could be explained by a different revascularization process which occurs in the graft [16,17,33]. A better blood perfusion may lead improved graft integration and less soft tissue resorption. The current literature review also identified two prospective studies reporting results of XCMs in per-implant soft tissue augmentation. Pompa and Papi [27], using a Mucoderm^®^ XCM indicated at 12 months post-surgery a mean PKMW gain of 4.32 ± 1.22 mm, while Schallhorn et al. [28] at 6 months, using a Mucograft^®^ XCM indicated a mean PST and PKMW gain of 2.2 ± 0.9 mm and 2.1 ± 1.0 mm, respectively.

Recently, collagen-based matrices derived from porcine dermis (PDXCMs) have been introduced in dentistry as a substitute for CTG in peri-implant plastic surgery. Due to the preservation of their mechanical stability these collagen matrices allow cells adhesion, and proliferation and blood vessel in-growth [34,35], developing into a fully functional tissue [36,37].

The literature search did not allow to identify randomized clinical trials comparing PDXCMs vs. CTGs. However, some prospective pilot cohort studies reported data on in peri-implant soft tissue augmentation using PDXCMs (Table 5).

**Table 5 materials-15-03937-t005:** General overview of the results reported by selected studies in which PDXCM was used.

Study	Sample	Study Design	Follow-Up Time	Keratinized Mucosa Width Gain (mm)	Soft Tissue Thickness Gain (mm)
Papi et al. [27]	12 patients	Prospective cohort study	12 months	N R	PDXCM: 1.25
Zafiropoulos et al. [38]	27 patients	Prospective, randomized examiner-blinded controlled clinical study	6 months	N R	PDXCM: 1.06
Stefanini et al. [39]	10 patients	Case series	12 months	PDXCM 0.65 ± 0.41	PDXCM: 1.2 ± 0.18
Papi and Pompa 12 [40]	12 patients	Prospective pilot cohort study	12 months	PDXCM: 4.32	N R
Schmitt et al. [41]	14 patients	Controlled clinical trial	6 months	N R	PDXCM: 0.30 ± 0.16
Verardi et al. [42]	24 patients 24 implants	Prospective study	6 months	PDXCM 1.33 ± 0.71	N R

Papi et al. [27] documented a PKMW mean increase of 6.51 and 5.67 mm at 5 months and at 1-year post-surgery, respectively. At the last follow-up control (12 months) a mean shrinkage of 29% was observed. In another prospective study the same authors performed soft tissue peri-implant augmentation combining the use of PDXMs with synthetic bone [43]. Both PKMW and PST showed an increase from pre-surgery to the first month, a decrease during the first 12 months, and a stabilization from 12 to 24 months. Compared to the baseline, after 24 months PST and PKMW gained 1.94 ± 0.05 mm and 1.60 ± 0.11 mm, respectively. Similar results on PST increase have been reported also by Zafiropoulous et al. [38], who using PDXCMs in 27 patients and found at 6 months after surgery a significant increase of 1.06 mm. Stefanini et al. [39] in 10 patients used a coronal advanced flap surgical technique combined with PDXCMs obtaining at 1-year post-surgery 1.2 ± 0.18 mm gain of PST. Papi & Pompa. [40], using digital linear and volumetric measurements reported in 12 patients treated with PDXCMs a volumetric gain in PST of 51,501 mm^3^ with a mean shrinkage of 23.31%. Comparative clinical data between CTGs and PDXCMs in peri-implant plastic surgery with a 3D analysis have been reported also by Schmitt et al. [41]. Authors at 6 months post-surgery found a mean volume gain of 19.56 ± 8.95 mm^3^ and 61.75 ± 52.69 mm^3^ for PDXCM and CTG, respectively, and a mean increase of PST of 0.30 ± 0.16 mm for PDXCM versus 0.80 ± 0.61 mm for CTG. Aragoneses et al. [42] evaluated in female white pigs (*Sus scrofa domestica*) the clinical and histological differences in peri-implant soft tissue volume augmentation using PDXCM vs. CTG. Three months post-surgery the mean volume increase was 1.53 mm, and it decreased 6 months later of 0.51 mm due to shrinkage and PDXCM resorption. At 45 days, the biopsies corresponding to PDXCMs documented a complete epithelial healing with the presence of a well keratinized layer. At 90 days all sites treated with PDXCM showed a matured keratinized squamous stratified epithelium sustained by a connective tissue with correctly organized collagen fibers and normal vascularization. However, at 90 days post-surgery, sites treated with CTG, compared to those treated with PDXCM, showed a statistically significant higher thickness (approximately 60%). The variability of the results reported by the over mentioned studies might depend by the different surgical techniques and materials used [44,45,46]. The use of a surgical bilaminar technique with the PDXCM placement under a split-thickness buccal flap could favor the blood perfusion of the matrix, which has been described as one of the factors affecting the result. Other factors influencing clinical outcomes might be also connected to the different suturing techniques and materials (i.e., the needle’s characteristics, bite size, suture position, location of knots tied, etc.), and early suture removal (<10 days). Another important consideration is that commercially available PDXCMs have a standard thick, while after the withdrawal, CTG can be prepared to the desired thickness. To overcome some of the above-mentioned limitations (Table 6) authors of the current paper present preliminary results of a RCT in which a new inlay technique named Guided Soft Tissue Regeneration (GSTR) performed using a new PDXCM (NovoMatrix^TM^; BioHorizons, Birmingham, AL, USA) has been compared to CTG. In Figure 1a, Figure 2a, Figure 3, Figure 4a, Figure 5a and Figure 6a is presented a clinical case treated with PDXCM and GSTR technique. In Figure 1b, Figure 2b, Figure 4b, Figure 5b and Figure 6b is presented a clinical case treated with CTG.

Preliminary in vitro studies showed that the proprietary tissue processing of NovoMatrix ^TM^, maintains tissue stability, allows a rapid blood vessel in-growth and fibroblast adhesion and proliferation with a minimal inflammatory response [47,48,49].

Unlike the other commercially available PDXCMs, this matrix is made up of a single layer that can be sectioned and superimposed to obtain the desired thickness. Moreover, the bilaminar technique used, with a split thickness flap, allows to keep periosteum and muscular insertion in order to maintain periosteal vascularization of the bone and to have soft tissue available to suture the matrix.

Limitations: The electronic search of the present critical review allowed to select only seven RCTs that compare the use of XMCs vs. CTGs in plastic peri-implant surgery, while it did not allow to identify RCTs comparing PDXCMs vs. CTGs. All the selected studies reported results of PKMW and PST, but no results are related to PSTH and PBT. Given the limitation of the peri-implant phenotypic parameters described and the limited number of samples, results should be interpreted with caution. Moreover, not all the surgical procedure were performed at the same time points. Another limitation can de due to the fact that all RCTs were supported by companies that produce the XCMs. Many studies do not report information about the inclusion of smoking participants. Tobacco smoking affects the healing potential of the periodontal tissues, and it could therefore represent a confounding factor [50]. This, and other confounding factors, such as periodontal phenotype, type of implant, type of abutment, etc., should be also standardized in future studies with the aim to reduce bias.

## 5. Conclusions

Results of the present literature review suggest that the use of XCMs and PDXCMs is effective in increasing PKMW and PST around dental implants. Nevertheless, further studies are required to evaluate the long-term outcomes. Furthermore, RCTs are also required to evaluate the effectiveness of PDXCMs compared to CTG.

## Figures and Tables

**Figure 1 materials-15-03937-f001:**
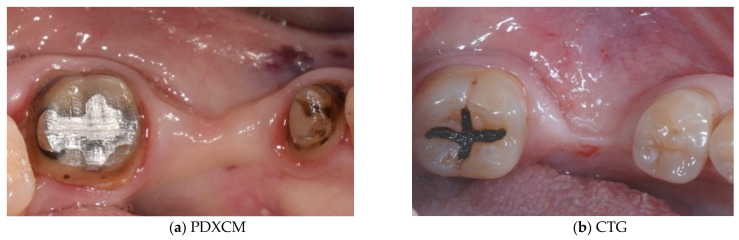
(**a**,**b**) Presurgical image of clinical case treated with PDXCM (right) and CTG (left).

**Figure 2 materials-15-03937-f002:**
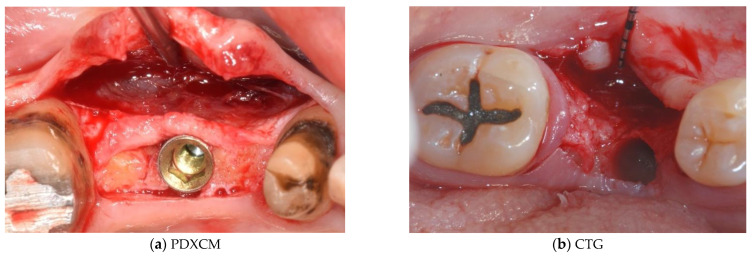
(**a**,**b**) with a bilaminar technique a split thickness flap allows us to keep periosteum and muscular insertion in order to maintain periosteal vascularization of the bone and to have soft tissue available to suture the matrix or CTG.

**Figure 3 materials-15-03937-f003:**
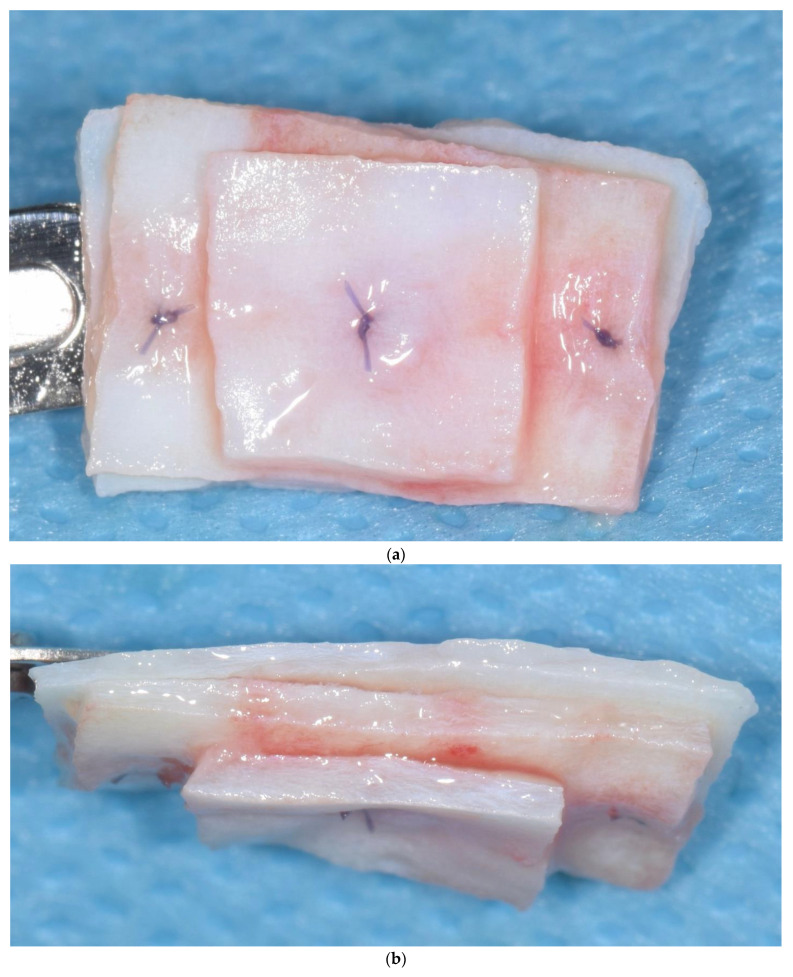
(**a**,**b**) A new PDXCM (NovoMatrix^TM^; BioHorizons, Birmingnam, AL, USA) was used, with an inlay technique. The PDXCM can be sectioned and aggregated in multiple layers sutured to each other in one unique inlay graft ready to be placed on the bleeding bed around the implant and fixed to the periosteum and/or to the flap.

**Figure 4 materials-15-03937-f004:**
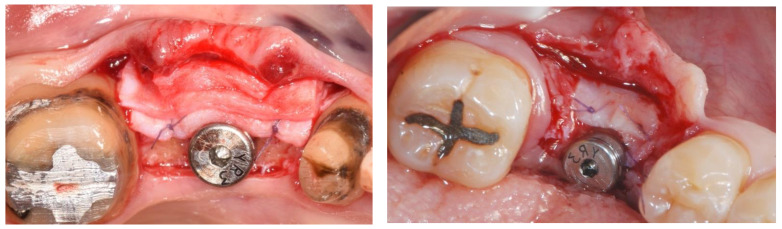
(**a**,**b**) The PDXCM is sutured to periosteum (right) and CTG is sutured to periosteum (left).

**Figure 5 materials-15-03937-f005:**
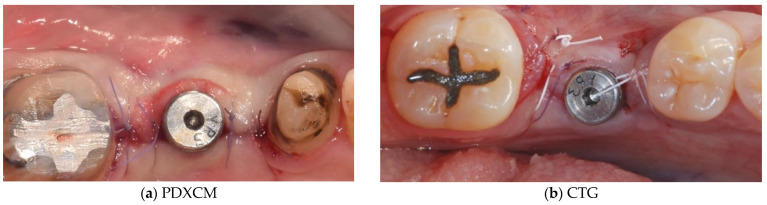
(**a**,**b**) the PDXCM is completely covered by the flap which ensures a double vascularization. The double vascularization is essential to guarantee to both side of the graft the opportunity of a quick integration with the surrounding tissue.

**Figure 6 materials-15-03937-f006:**
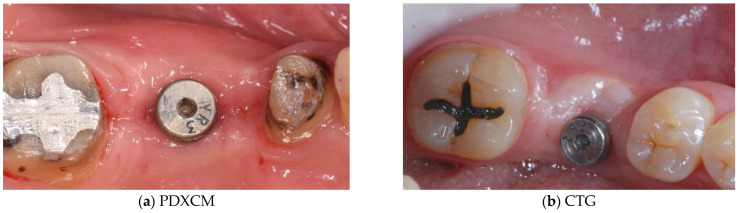
(**a**,**b**) Clinical situation after 3 months.

**Table 1 materials-15-03937-t001:** PRISMA flow chart.

Identification of Studies via Databases and Registers
**Identification**	Records identified fromPubMed searching: (*n* = 87)	Records identified fromScopus searching: (*n* = 77)	Records identified from Cochrane Library: (*n* = 75)	Records identified from Web of Sciences: (*n* = 5)
	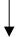	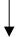	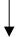	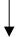
**Screening**	Records after duplicates removed (*n* = 133)
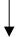		
Records screened (*n* = 133)	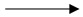	Records excluded by title and abstract (*n* = 122)
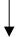		
Full text assessed for eligibility	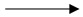	Full text excluded after full text assessed (*n* = 4) Reasons for exclusion: no comparison between CTG and XCM
	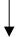		
Included	Studies included in review (*n* = 7)	

**Table 2 materials-15-03937-t002:** General overview of the included RCT which compared CTG (control) versus XCM (test).

General Overview of RCT Which Compared XCMs Versus CTGs						
Study	Follow-Up	Patients/Implants	Systemic Periodontal Status Smoking	Time of Surgery	Outcomes Measurements	XCM
Sanz et al. [20]	1,3,6 months	P = 14 I = 14	Systemic, Periodontally healthy FMPI < 20% Smokers < 10 sig.die	After crown placement	PKMW, PPD, CAL, GI, PI, pain, PAS	Mucograft^®^
Lorenzo et al. [21]	6 months	P = 24 I = 24	Systemic, Periodontally healthy FMPI < 20% Smokers < 10 sig.die	After crown placement	PKMW, GI, PI, PD, CAL	Mucograft^®^
Thoma et al. [22]	3 months	P = 20 I = 20	Systemic, Periodontally healthy FMPI < 20% Smokers < 10 sig.die	After implant placement. From 6 weeks to 6 months before	PKMM, PPD, CAL, BOP, PI	Mucograft^®^
Zeitner et al. [23]	3 months	P = 20 I = 20	Systemic, Periodontally healthy FMPI < 20% Smokers < 10 sig.die	After implant placement. From 6 weeks to 6 months before	PKMM, PPD, CAL, BOP, PI	Mucograft^®^
Cairo et al. [24]	6 months	P = 60 I = 60	Systemic, Periodontally healthy FMPI < 15% PPD < 5mm Smokers < 10 sig.die	During second surgery implant uncovering	PKMW, GT, PD, PAS	Mucograft^®^
Puzio et al. [25]	12 months	P = 22 I = 30	Systemic, Periodontally healthy PI < 20% FMBS < 15% Smokers < 10 sig.die	During second surgery implant uncovering	PKMW, GT	Mucograft^®^
Huber et al. [26]	12 months	P = 20 I = 20	Systemic, Periodontally healthy Smokers < 10 sig.die	During second surgery implant uncovering	PKMW, GT	Mucograft^®^
**General overview of prospective studies which investigated XCMs**						
Pompa & Papi. [27]	12 months	P = 12 I = 10	Systemic, Periodontally healthy FMPI < 20% Smokers < 10 sig.die	After crown placement	KMW, PI, PD, BP	Mucoderm^®^
Schallhorn et al. [28]	6 months	P = 30 I = 32	Systemic, Periodontally healthy FMPI < 20% Smokers < 10 sig.die	After crown placement	KMW, GT, PD, colour, PAS	Mucograft^®^

**Table 3 materials-15-03937-t003:** Assessments of the risk of bias.

Study	Adequate Sequence Generation	Allocation Concealment	Blinding	Incomplete Outcomes Data Addressed	Selective Outcome Reporting	Free of Other Source of Bias	Estimate Potential Source of Bias
Sanz et al. [20]	No	Yes	Yes	Yes	Yes	Yes	Low risk
Lorenzo et al. [21]	Yes	Yes	Yes	Yes	Yes	Yes	Low risk
Thoma et al. [22]	Yes	Yes	Yes	Yes	Yes	Yes	Low risk
Zeitner et al. [23]	Yes	Yes	Yes	Yes	Yes	Yes	Low risk
Cairo et al. [24]	Yes	Yes	Yes	Yes	Yes	Yes	Low risk
Puzio et al. [25]	Yes	Yes	Yes	Yes	Yes	Yes	Low risk
Huber et al. [26]	Yes	Yes	Yes	Yes	Yes	Yes	Low risk

**Table 6 materials-15-03937-t006:** Advantages/disadvantages of XCM, PDXCM, vs. CTG.

	Advantages	Disadvantages
CTG	- low shrinkage after the healing period - is completely incorporated histologically - more effective in generating attached tissue	- the palate is healed by secondary intention and requires a dressing for 10 to 14 days, which is uncomfortable for most patients - inability to harvest large grafts, - high morbidity rates after surgery, - poor aesthetics due to differences in texture and color from adjacent areas. - High risk of complications
XCMs/PDXCMs	- do not need a donor site - provide better aesthetic results. - no complication if exposed - no dimensional limits of withdrawal - patients in group reported having experienced significantly less pain until 7 days	- great shrinkage after the healing period - is not completely incorporated histologically - less effective in generating attached tissue

## Data Availability

Not applicable.

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
