# Peer review of "Use of Xenogenic Collagen Matrices in Peri-Implant Soft Tissue Volume Augmentation: A Critical Review on the Current Evidence and New Technique Presentation"

_materials, 2022, doi:10.3390/ma15113937_

Round 1
Reviewer 1 Report
Serious errors are observed in the handling of bibliographical references, which discredit the article (which, on the other hand, is well prepared) and force it to be rejected unless the errors detected are corrected in detail:
- From reference 3 to 5, where is 4?
- from reference 36 it passes to 38 and 37 appears later. You have to keep order of appearance.
- The same thing happens with reference 40, which comes out before 39 and then JUMPS to 46 (which is also wrong, it's not Schmitt)
- Bad order also in the 48-47-49
- The references 41-42-43-44-45 that appear at the end are missing from their site.
It is necessary to relocate the references and renumber CAREFULLY.
Is reference 40 in table 3 correct?
There is a change in line spacing starting from reference 35. Reference 36 has no volume or pages.
Reference 37 is not formatted correctly
Same as 38-39-40-41-42 and 49 In short,
serious errors in the bibliography and its citations in the text that cannot be accepted.
Author Response
Serious errors are observed in the handling of bibliographical references, which discredit the article (which, on the other hand, is well prepared) and force it to be rejected unless the errors detected are corrected in detail:
- From reference 3 to 5, where is 4?
- from reference 36 it passes to 38 and 37 appears later. You have to keep order of appearance.
- The same thing happens with reference 40, which comes out before 39 and then JUMPS to 46 (which is also wrong, it's not Schmitt)
Response: The reference 41, 42, 43, 44, 45, 47 have been not reported in the table because
- Bad order also in the 48-47-49
- The references 41-42-43-44-45 that appear at the end are missing from their site.
It is necessary to relocate the references and renumber CAREFULLY.
Is reference 40 in table 3 correct?
There is a change in line spacing starting from reference 35. Reference 36 has no volume or pages.
Reference 37 is not formatted correctly
Same as 38-39-40-41-42 and 49 In short,
serious errors in the bibliography and its citations in the text that cannot be accepted.
Response: in the revised manuscript all references have been carefully renumbered

Reviewer 2 Report
Dear authors,
The issued addressed in your manuscript about the potential of xenogeneic collagen matrices compared with autologous connective tissue graft in peri-implant tissue volume augmentation is a very interesting and important topic that falls within the scope of Materials.
According to my peer-review, the present manuscript may be accepted for publication only after a major review covering the following aspects:
- Please clarify the nature of your review. Title indicates that a literature review was performed. But, what type? In Materials and Methods, aspects of a systematic review were applied, including the expression of a PRISMA flowchart and an assessment of risk of bias. Authors are encouraged to reformulate and improve this research work to a systematic review.
- In addition, I disagree that any other type of study should be included in the manuscript (such a case presentation or as authors refer in Discussion "present preliminary results of RCT in 275 which a new inlay technique performed with a new PXDM has been compared to CTG. Therefore, it is confusing that two type of studies are intended to be presented in the same manuscript.
- By reformulating the present research to a systematic review, your aim should be more specific (line 60 to 63) and followed by a PICO question.
-
Was the PRISMA statement accomplished by the authors?
- I reiterate the importance of providing a systematic review to be in light with the word chosen by authors "Effectiveness". I agree it is appropriated. But, how should authors measure "effectiveness"? And what was the method presented in this manuscript?
- Why do authors refer that "peri-implant biologic width" is an outdated term? (line 45 to 47). I do not agree. Which literature supports this perspective?
- It is a little bit unconvincing that after quality assessment using the Cochrane Collaboration's tool, all the randomised clinical trials included provide the same score. Please review this aspect and provide an appropriate discussion about this finding.
- Limitations of the included studies should be better identified and expressed in the Discussion;
- English editing is suggested through the manuscript text.
Author Response
Dear authors,
The issued addressed in your manuscript about the potential of xenogeneic collagen matrices compared with autologous connective tissue graft in peri-implant tissue volume augmentation is a very interesting and important topic that falls within the scope of Materials.
According to my peer-review, the present manuscript may be accepted for publication only after a major review covering the following aspects:
- Please clarify the nature of your review. Title indicates that a literature review was performed. But, what type? In Materials and Methods, aspects of a systematic review were applied, including the expression of a PRISMA flowchart and an assessment of risk of bias. Authors are encouraged to reformulate and improve this research work to a systematic review.
- Response: The manuscript does not have the characteristics of systematic reviews that are already present in the literature (Gargallo-Albiol J, Barootchi S, Tavelli L, Wang HL. Efficacy of Xenogeneic Collagen Matrix to Augment Peri-Implant Soft Tissue Thickness Compared to Autogenous Connective Tissue Graft: A Systematic Review and Meta-Analysis. Int J Oral Maxillofac Implants. 2019 September / October; 34 (5): 1059-1069.Moraschini V, Guimarães HB, Cavalcante IC, Calasans-Maia MD. Clinical efficacy of xenogeneic collagen matrix in augmenting keratinized mucosa round dental implants: a systematic review and meta-analysis. Clin Oral Investig. 2020 Jul; 24 (7): 2163-2174).
The purpose of the manuscript is to provide a critical review on the current evidence (as added in the text) on the use of XCMs in plastic peri-implant surgery, as in recent years there has been an increasing number of matrices available on the market
- In addition, I disagree that any other type of study should be included in the manuscript (such a case presentation or as authors refer in Discussion "present preliminary results of RCT in 275 which a new inlay technique performed with a new PXDM has been compared to CTG. Therefore, it is confusing that two type of studies are intended to be presented in the same manuscript.
Response: the intention of the authors is to provide the reader an update and a critical analysis of what has been also published on use of PXCMs (RCT, Longitudinal studies, Case Series studies) in plastic peri-implant surgery and to present preliminary results on the use of a new xenogenic matrix.
For this reason the title has been changed in: “Use of xenogenic collagen matrices in peri-implant soft tissue volume augmentation: a critical review on the current evidence and new technique presentation”
- By reformulating the present research to a systematic review, your aim should be more specific (line 60 to 63) and followed by a PICO question.
In the revised manuscript, Pico questions were added: Focused questions (based on PICO criteria : What are the clinical effects of XCM (I) relative to different soft tissue augmentation methods (C) on improving PKMW, PST and PSTH (O) around dental implants (P)?
- Was the PRISMA statement accomplished by the authors?
Response: In the revised manuscript it has been added: “In order to increase the quality and transparency of the study, the PRISMA checklist was followed.”
- I reiterate the importance of providing a systematic review to be in light with the word chosen by authors "Effectiveness". I agree it is appropriated. But, how should authors measure "effectiveness"? And what was the method presented in this manuscript?
- Response: The manuscript does not have the characteristics of systematic reviews that are already present in the literature (Gargallo-Albiol J, Barootchi S, Tavelli L, Wang HL. Efficacy of Xenogeneic Collagen Matrix to Augment Peri-Implant Soft Tissue Thickness Compared to Autogenous Connective Tissue Graft: A Systematic Review and Meta-Analysis. Int J Oral Maxillofac Implants. 2019 September / October; 34 (5): 1059-1069.Moraschini V, Guimarães HB, Cavalcante IC, Calasans-Maia MD. Clinical efficacy of xenogeneic collagen matrix in augmenting keratinized mucosa round dental implants: a systematic review and meta-analysis. Clin Oral Investig. 2020 Jul; 24 (7): 2163-2174).
The purpose of the manuscript is to provide a critical review on the current evidence (as added in the text) as in recent years there has been an increasing number of matrices available on the market.
According to the suggestion of the reviewer, the title of the revised manusctipt has been changed” Use of xenogenic collagen matrices in peri-implant soft tissue volume augmentation: a critical review on the current evidence and new technique presentation”
- Why do authors refer that "peri-implant biologic width" is an outdated term? (line 45 to 47). I do not agree. Which literature supports this perspective?
Response: The term “peri-implant biologic width” has been recently discussed in a paper: Jepsen S, Caton JG, et al. Periodontal manifestations of systemic diseases and developmental and acquired conditions: Consensus report of workgroup 3 of the 2017 World Workshop on the Classification of Periodontal and Peri-Implant Diseases and Conditions. J Periodontol. 2018;89(Suppl 1):S237–S248.
In the paper it has been reported: “What is the biologic width? Biologic width is a commonly used clinical term to describe the apico-coronal variable dimensions of the supracrestal attached tissues. The supracrestal attached tissues are histologically composed of the junctional epithelium and supracrestal connective tissue attachment. The term biologic width should be replaced by supracrestal tissue attachment. (pg. 245).
- It is a little bit unconvincing that after quality assessment using the Cochrane Collaboration's tool, all the randomised clinical trials included provide the same score. Please review this aspect and provide an appropriate discussion about this finding.
Response: in Table 3 is reported the “Assessments of the risk of bias”. As listed, based on the selected parameters, all RCTs had a low risk of bias. However in “Discussion” of the revised manuscript, “limitation” have been discussed.
- Limitations of the included studies should be better identified and expressed in the Discussion;
Response: in “Discussion” of the revised manuscript limitations of the included studies have been reported: Limitations: The present critical review includes only seven RCTs that compare the use of XMC vs. CTG in plastic peri-implant surgery. Given the limited number of samples, results should be interpreted with caution. Moreover, not all the surgical procedure were performed at the same time points. Another limitation can de due to the fact that all RCTs were supported by companies that produce the XCMs. Many studies did not report information about the inclusion of smoking participants. Tobacco smoking affects the oral environment and ecology, the vascularization of the gingival tissues, the immune and inflammatory responses and the healing potential of the periodontal connective tissues [50]. Tobacco smoking and other confounding factors, such as periodontal phenotype, type of implant, type of abutment, etc., should be also standardized in future studies with the aim to reduce bias.
- English editing is suggested through the manuscript text.
Response: the revised manuscript has been correct by a native English speaking person

Reviewer 3 Report
The authors have presented an in depth literature review on the Effectiveness of xenogenic collagen matrices vs. autologous connective tissue grafts in peri-implant soft tissue volume augmentation. Critical review of the use of XCM in increasing PKMW and PST around dental implants from literature were comprehensively reported. I would recommend this review paper to be published.
Author Response
This is a well-written systematic review on an interesting topic.
The authors should better stress in the introduction the originality of the study.
Response: in the “Introduction” of the revised manuscript the originality of the study has been better stress.
“XCM is a biomaterial composed of collagens type I and type III. Collagen has angiogenic function, which accelerates healing and enhances fibroblast expression. In addition, because it is enzymatically biodegradable, it does not need to be removed [16]. The collagen arrangement within XCMs is organized with pores that allow vascularization and provide a framework for connective tissue cell migration [16]. In addition, the matrix thickness acts as a space maintainer favoring the formation of keratinized tissue [16]. Thanks to all these features, XCM could be an alternative for the increase of peri-implant soft tissue in replacement of the autogenous grafts. Several RCTs have investigated the clinical outcome of XCMs in plastic peri-implant surgery]. These studies have shown favorable clinical results; however, the available data demonstrate that surgical outcomes might be influenced by many conditions. In this context, the aim of this literature review is to investigate whether the use of XCMs provides similar outcomes of CTGs in peri-implant soft tissue augmentation procedures and to critically analyze the available evidence. Moreover, a clinical case with a new XCM and a new grafting procedure is presented.
A table or figure detailing the advantages/ disadvantages, indications/ contraindications of each material would be useful.
Response: in the revised manuscript, a new table (n.4) reporting advantages/ disadvantages has been added.
Reviewer 4 Report
The manuscript is aimed to investigate whether the use of xenogenic collagen matrices (XCM) provides similar outcomes of autologous connective tissue grafts (CTG) in peri-implant soft tissue volume augmentation. This manuscript is interesting and especially describes the morphologic and dimensional features of soft tissue components around dental implants. However, several doubts and questions should be clarified as the followings:
- In abstract: “The electronic search provided a total of 135 articles”, however, in Table 1. PRISMA flow chart: “Records after duplicates removed (n. 133). Which one is correct?
- In Introduction, the authors wrote: “The morphologic and dimensional features of soft tissue components around dental implants, includes 4 parameters: 1) the peri-implant keratinized mucosa width (PKMW), 2) the peri-implant soft tissue thickness (PST), 3) the peri-implant supra-crestal tissue height (PSTH), and 4) the peri-implant bone thickness (PBT)”, however, only two parameters that are mention in the results and discussion: PKMW and PST. Why? The two other parameters that are not mention in results should be discussed in Discussion, PSTH and PBT.
- Table 2. General overview of the included RCT which compared CTG (control) versus XCM (test) à Is there any specific reason why only Mucograft® used in this review? (There are other XCM, such as Fibro-Gide and Mucoderm).
- Table 3, the authors wrote PDXM, PADM, but their abbreviations cannot be found. Also in discussion, abbreviation for PXDM (page 9 line 276 and page 12 line 313) and PDXCs in conclusion cannot be found.
- Table 4. General overview of the results reported by selected RCT which compared CTG (control) versus XCM (test), the changes in PKMW between baseline and final follow up can be defined:
- a) XCM has similar results with CTG: 3 studies (Sanz, Lorenzo, Zeitner)
- b) XCM was more effective than CTG: 2 studies (Thoma, Cairo)
- c) CTG was more effective than XCM: 2 studies (Huber, Puzio)
Based on the Table 4, the XCM and CTG has the similar results, but, why the results of the present review suggest that the use of XCM is effective in increasing PKMW and PST around dental implants?
- This review is limited in to the quite small number of studies (7 studies) and there are many reviews that analyse the soft tissue augmentation in different clinical situations using comparison between XCM and CTG, what is your significant differential gain?
- What is the limitation of your review? Please write them in your manuscript.
Author Response
The manuscript is aimed to investigate whether the use of xenogenic collagen matrices (XCM) provides similar outcomes of autologous connective tissue grafts (CTG) in peri-implant soft tissue volume augmentation. This manuscript is interesting and especially describes the morphologic and dimensional features of soft tissue components around dental implants. However, several doubts and questions should be clarified as the followings:
- In abstract: “The electronic search provided a total of 135 articles”, however, in Table 1. PRISMA flow chart: “Records after duplicates removed (n. 133). Which one is correct?
Response: in the Abstracts of the revised manuscript the number “135” has been corrected with”133”
- In Introduction, the authors wrote: “The morphologic and dimensional features of soft tissue components around dental implants, includes 4 parameters: 1) the peri-implant keratinized mucosa width (PKMW), 2) the peri-implant soft tissue thickness (PST), 3) the peri-implant supra-crestal tissue height (PSTH), and 4) the peri-implant bone thickness (PBT)”, however, only two parameters that are mention in the results and discussion: PKMW and PST. Why? The two other parameters that are not mention in results should be discussed in Discussion, PSTH and PBT.
Response: in “Discussion” of the revised manuscript PSTH and PBT have been mentioned. “None of the studies selected by the current review report results relating to PSTH and PBT after treatment with XCMs vs. CTG. In peri-implant soft tissue augmentation procedures, an increase in PSTH should still be within the therapeutic goals as it has been shown that a thickness > 2mm could have a protective effect on peri-implant marginal bone resorption. In addition, even a minimum thickness of 1.5 mm of PST should always be present to reduce peri-implant bone remodeling.
- Table 2. General overview of the included RCT which compared CTG (control) versus XCM (test) à Is there any specific reason why only Mucograft® used in this review? (There are other XCM, such as Fibro-Gide and Mucoderm).
Response: No RCTs are present in literature comparing Fibro-Gide and Mucoderm vs. CTG. In “Discussion” and in Tables are reported results of 2 prospective studies in which others matrices were used.
- Table 3, the authors wrote PDXM, PADM, but their abbreviations cannot be found. Also in discussion, abbreviation for PXDM (page 9 line 276 and page 12 line 313) and PDXCs in conclusion cannot be found.
Response: in the revised manuscript all acronyms have been spelled correctly
- Table 4. General overview of the results reported by selected RCT which compared CTG (control) versus XCM (test), the changes in PKMW between baseline and final follow up can be defined:
- a) XCM has similar results with CTG: 3 studies (Sanz, Lorenzo, Zeitner)
- b) XCM was more effective than CTG: 2 studies (Thoma, Cairo)
- c) CTG was more effective than XCM: 2 studies (Huber, Puzio)
Response: in the revised manuscript in Table outcomes have been better defined
- Based on the Table 4, the XCM and CTG has the similar results, but, why the63 results of the present review suggest that the use of XCM is effective in increasing PKMW and PST around dental implants?
Response: in the revised manuscript in Table outcomes have been better defined
- This review is limited in to the quite small number of studies (7 studies) and there are many reviews that analyse the soft tissue augmentation in different clinical situations using comparison between XCM and CTG, what is your significant differential gain?
Response: The differential gain of the present paper is to present a critical review on the current evidence and new technique presentation. Several studies have shown favorable clinical results; however, the available data demonstrate that surgical outcomes might be influenced by many conditions. In this context, the aim of this literature review is to investigate whether the use of XCMs provides similar outcomes of CTGs in peri-implant soft tissue augmentation procedures and to critically analyze the available evidence.
- What is the limitation of your review? Please write them in your manuscript.
Response: In the revised manuscript it has been added:
Limitations: The electronic search of the present critical review allowed to select only seven RCTs that compare the use of XMCs vs. CTGs in plastic peri-implant surgery, while it did not allow to identify RCTs comparing PDXCMs vs. CTGs. All the selected studies reported results of PKMW and PST, but no results are related to PSTH and PBT. Given the limitation of the peri-implant phenotypic parameters described and the limited number of samples, results should be interpreted with caution. Moreover, not all the surgical procedure were performed at the same time points. Another limitation can de due to the fact that all RCTs were supported by companies that produce the XCMs. Many studies do not report information about the inclusion of smoking participants. Tobacco smoking affects the healing potential of the periodontal tissues, and it could therefore represent a confounding factor [50]. This, and other confounding factors, such as periodontal phenotype, type of implant, type of abutment, etc., should be also standardized in future studies with the aim to reduce bias.

Reviewer 5 Report
This is a well-written systematic review on an interesting topic.
The authors should better stress in the introduction the originality of the study.
A table or figure detailing the advantages/ disadvantages, indications/ contraindications of each material would be useful.
Author Response
This is a well-written systematic review on an interesting topic.
The authors should better stress in the introduction the originality of the study.
Response: in the “Introduction” of the revised manuscript the originality of the study has been better stress.
“XCM is a biomaterial composed of collagens type I and type III. Collagen has angiogenic function, which accelerates healing and enhances fibroblast expression. In addition, because it is enzymatically biodegradable, it does not need to be removed [16]. The collagen arrangement within XCMs is organized with pores that allow vascularization and provide a framework for connective tissue cell migration [16]. In addition, the matrix thickness acts as a space maintainer favoring the formation of keratinized tissue [16]. Thanks to all these features, XCM could be an alternative for the increase of peri-implant soft tissue in replacement of the autogenous grafts. Several RCTs have investigated the clinical outcome of XCMs in plastic peri-implant surgery]. These studies have shown favorable clinical results; however, the available data demonstrate that surgical outcomes might be influenced by many conditions. In this context, the aim of this literature review is to investigate whether the use of XCMs provides similar outcomes of CTGs in peri-implant soft tissue augmentation procedures and to critically analyze the available evidence. Moreover, a clinical case with a new XCM and a new grafting procedure is presented.
A table or figure detailing the advantages/ disadvantages, indications/ contraindications of each material would be useful.

Round 2
Reviewer 2 Report
Dear authors,
Major review questions have been clearly answered in you review.
Congratulations. Minor english editing is suggested before publications.
kind regards,
This manuscript is a resubmission of an earlier submission. The following is a list of the peer review reports and author responses from that submission.